# Journey or Destination? Rethinking Pilgrimage in the Western Tradition

Anne E. Bailey

Faculty of History, University of Oxford, Oxford OX1 2RL, UK; anne.bailey@history.ox.ac.uk

**Abstract:** Pilgrimage is undergoing a revival in western Europe, mainly as newly established or revitalised pilgrim routes, such as the Camino de Santiago in northern Spain. These trails have helped to foster the widespread idea that pilgrimage is essentially a journey: a spiritual or "meaningful" journey undertaken slowly, and preferably on foot, in the medieval tradition. The purpose of this article is to problematise this journey-oriented understanding of pilgrimage in Christian and post-Christian societies and to suggest that the importance given to the pilgrimage journey by many scholars, and by wider society, is more a product of modern Western values and post-Reformation culture than a reflection of historical and current-day religious practices. Drawing on evidence from a range of contemporary sources, it shows that many medieval pilgrims understood pilgrimage as a destination-based activity as is still the case at numerous Roman Catholic shrines today.

**Keywords:** pilgrimage; medieval; Christianity; Camino de Santiago; Holy Land pilgrimage; sacred journeys; shrines; Roman Catholicism; Medievalism

## 1. Introduction

Few academic topics are as expansive and dynamic as pilgrimage which, over the past thirty or forty years, has embraced an array of ritual practices across great swathes of time and cultures. Attempting to pin down any one definition is fraught with difficulties. However, most academics broadly understand pilgrimage as a spiritual endeavour incorporating two essential elements: a journey and a sacred destination. In an attempt to distinguish pilgrimages from more mundane visits to holy places, some scholars have gone further in suggesting that the journey component should be "religious" or "spiritual" in its undertaking and not simply a method of travelling from A to B (Raj et al. 2015; LeSueur 2018, p. 16; Goodnow and Bloom 2017).

While both the journey and the destination elements of pilgrimage have received much attention across various disciplines in the Western academic world, there has been a recent tendency among historians and social scientists to emphasise the journey over the destination, particularly in Western contexts. A similar leaning is evident in popular culture, as recognised by the author of a *Sunday Telegraph* article voicing the popular sentiment that, "It's the journey rather than the destination that matters to today's pilgrims". Referring to the Middle Ages she adds, "perhaps it was always thus" (Simmons 2019).

This article discusses this shift in perspective and argues that the importance so often given to journeying in academia and in wider society is more a product of Western values, preconceptions, and ideals than a reflection of historical and current-day religious practices. The article also challenges the related idea, especially prevalent among modern pilgrims walking Europe's revitalised pilgrim trails, that "only a journey by foot" constitutes a "proper" pilgrimage (Welch 2019). It illustrates how the projection of contemporary worldviews back into the past has led to many erroneous preconceptions about medieval pilgrimage, even prompting the notion that Chaucer's pilgrims travelled to Canterbury on foot (Cash 2020; Kelly and Kelly 2018, p. 3; Gogerty 2019).

The pilgrimage which best exemplifies modern suppositions about pedestrian travel is the Camino de Santiago, the web of dedicated pilgrim trails across northern Europe

which converge at the shrine of St James at Santiago de Compostela in northwest Spain. Pilgrims have been travelling to Santiago since the ninth century, when an enterprising bishop announced the surprise discovery of the relics of St James the Great. So successful was the promotion of the Apostle's cult that, within two centuries, Santiago had become the third most important Christian pilgrimage site after Jerusalem and Rome. Although the pilgrimage went into decline in the Early Modern period, its remarkable recovery in the late twentieth century has been an important impetus for the current pilgrimage revival. In 2022, 438,683 pilgrims were recorded reaching Santiago (Oficina del Peregrino). The majority of these (95%) undertook their Camino on foot, the method considered most historically authentic.

One of the Camino's attractions for modern walkers is the conviction that they are continuing an ancient custom by travelling at "medieval speed", "as the original pilgrims did" (Genoni 2010, pp. 157–75, 165; Frey 1998, p. 41; Nilsson and Tesfahuney 2016, p. 24), a concept vigorously promoted by the Council of Europe and the Roman Catholic Church (Gardner et al. 2016). The idea that "true" pilgrimage to Santiago involves a long, drawn-out journey on foot is given added weight by the certificates of completion, known as "Compostelas", awarded to those who spurn mechanised transport and walk the last hundred kilometres. In the twenty-first century, the journey is typically perceived as the main point of the endeavour (Welch 2009; Peelan and Jansen 2007). For many Camino pilgrims, their ultimate destination—the shrine of St James—simply marks the finishing line and may even be considered an anti-climax (Nilsson and Tesfahuney 2016). For one retired Anglican minister, it was simply a place to reflect on the journey (Gibbs 2017, pp. 22–24).

The hardships endured by modern pilgrims unused to long-distance walking are commonly assumed to be analogous to the travel experiences of pilgrims in earlier times, for whom walking was a penitential activity which promised spiritual rewards (Genoni 2010; Ron and Timothy 2018, p. 3). Medieval sources, however, reveal a more nuanced reality. They show that long-distance, penitential walking was by no means the only way of experiencing pilgrimage. For many, the journey to a saint's shrine seems to have had little or no spiritual value, and it was only when their destination was reached that penitential exercises were performed. As for the pilgrimage to Santiago, it remained largely a destination-based phenomenon until the 1970s, when the medieval route was promoted as a cultural tourist itinerary, and walkers began joining those travelling to the city's cathedral by car (Talbot 2016).

Drawing on medieval sources, on modern ethnographical studies, and on the author's personal observations and experiences, this article offers a corrective to the increasingly popular idea in the secular West that pilgrimage should be understood chiefly as a journey. Focusing on voluntary pilgrimages to sacred shrines[1], it argues that the "journeying" model of pilgrimage often overlooks other forms of the practice, including the numerous destination-based pilgrimages practised by contemporary Roman Catholics and their medieval forebears.

## 2. Pilgrimage in Modern Scholarship and Post-Secular Culture

The emphasis on the pilgrimage journey over the destination is common among anthropologists and medievalists alike. Representing the latter, Thomas Head describes medieval pilgrimage as, "the process by which pious Christians come to the shrines of the saints" (Head 2001, p. 275), while Linda Kay Davidson and Maryjane Dunn-Wood understand the practice as, "The physical journey from one's normal place of residence to a religious shrine" (Davidson and Dunn-Wood 1993, p. 13). Some medievalists seemingly ignore the destination component completely. Nicholas Orme defines pilgrimage as, "A religious journey which is not part of the routine of life" (Orme 2018, p. 6).

Anthropologists' definitions are often similarly journey orientated. Antón Pazos refers to modern pilgrimage as, "a journey, especially a long journey to some sacred place or shrine undertaken as act of devotion" (Pazos 2012, p. 1). Many definitions go so far as to give the impression that pilgrimage primarily, if not exclusively, centres on the journey. We

find, for example, that pilgrimage is, "a religious journey made out of faith or devotion" (Frey 1998, p. 26), "a traditional religious or modern secular journey" (Collins-Kreiner 2010, p. 440), and "a substantial journey undertaken with reverential intent" (LeSueur 2018, p. 21). In his edited volume, *Sacred Journeys*, Alan Morinis explains that, "journey deserves primary attention because it is the essence of pilgrimage" (Morinis 1992, p. 17). While this slant towards journeying is by no means universal, its strengthening in Western academia might be illustrated by the recent publication of volumes dedicated to the study of pilgrimage routes (Olsen and Trono 2018; Olsen et al. 2023).

The current emphasis on pilgrimage travel can be traced back to a sea change in Anglophone scholarship in the late 1980s and early 1990s. Prior to this time, pilgrimage studies tended to focus on religious sites and their attendant rituals, and the prevailing view among scholars in the 1970s and 1980s was that saints' shrines were "cult centres" which drew in pilgrims as a means of maintaining social cohesion. Influenced in part by the "sacred centre" hypothesis of Mircea Eliade, the cult-centre model was popularised, among others, by Victor and Edith Turner (Turner and Turner 1978; Turner 1973) and Erik Cohen (1992), as well as by medievalists researching local saints' cults, such as Peter Brown (1981) and Ronald Finucane (1977).

A challenge to this interpretative model came first in the social sciences with the revisionist pilgrimage anthropology of the 1990s, and most notably with the publication of Simon Coleman's and John Eade's *Reframing Pilgrimage: Cultures in Motion* (Coleman and Eade 2004) which, like Eade and Sallnow's (1991) *Contesting the Sacred*, set itself up as a corrective to Victor Turner's place-centred approach. From the 1990s onwards, pilgrim shrines were less likely to be perceived as sacred centres studied in isolation from one another and were increasingly understood as interconnected locales situated within a fluid network of social interactions and cultural interchange. As a consequence, pilgrim travel—rather than pilgrimage destinations—rose to the forefront of scholarly interest. *Reframing Pilgrimage* came at a time when the theme of mobility was in vogue, and it was followed by a flood of studies linking pilgrimage with migration, globalisation, and other topical forms of movement, such as commodity flows, cultural exchange, and the emotions (Maddrell et al. 2016; Hyndman-Rizk 2012; Eade and Katič 2014; Jansen and Notermans 2012; Pazos 2013). Journeying clearly spoke to the zeitgeist of the time in a way that static destinations did not.

Although medievalists focusing on long-haul pilgrimage have tended to take a more balanced approach to the subject—with Jonathon Sumption (2002) and Diana Webb (2000), for example, giving equal weight to the "journey" and "destination" aspects—those researching local pilgrimage shrines have recently begun to forge a path from "destination" to "journey" similar to that of social scientists. Here, the shift in thinking seems to have been influenced, in part, by the rise to prominence of landscape archaeology in the 1980s, which sought to locate prehistorical, and historical, sites within wider geographical, social, and religious contexts.

Tracing historic routes across landscapes between sacred sites had obvious attractions for British medievalists with so much of their cultural heritage still visible on the ground, and some began casting their gaze beyond particular cult centres to wider pilgrimage networks. At the turn of the millennium, an interdisciplinary approach to pilgrimage studies brought archaeology and history into a more amicable relationship than in previous years, and fruitful cross-fertilisation inspired research projects, such as Martin Locker's survey, *Landscapes of Pilgrimage in Medieval Britain* (Locker 2015), which plotted four medieval pilgrim routes across the country. Other publications include Nicholas Orme's *Medieval Pilgrimage: With a Survey of Cornwall, Devon, Dorset, Somerset, and Bristol* (Orme 2018) and Emma Wells's (2016) *Pilgrim Routes of the British Isles* (2016). Collaborating with the tourism sector, medievalists have also helped to create a heritage route from Swansea to Hereford Cathedral based on the journey of thirteenth-century pilgrims (Clarke 2020).

Scholars do not, of course, work in cultural isolation, and the academic interest in pilgrimage routes and journeys reflects the growing public appetite for walking trails



with a pilgrimage theme. Pilgrimage trails have developed as part of a more general trend for cultural linear tourism, and their popularity seems to have much to do with modern Western sentiments and values, such as health and wellness, slow travel, eco-consciousness, the appeal of cultural heritage, and the desire for immersive, authentic experiences. However, as a form of pilgrimage which privileges journeys over destinations, walking pilgrimages sit particularly well within the Protestant cultural tradition, where the concept of pilgrimage is primarily understood as a journey or, more specifically, as a spiritual life journey (Bailey 2023). Walking pilgrimages conveniently take attention away from the destination traditionally associated in Protestant minds with theologically difficult issues, such as the veneration of saints and relics.

Over the last forty or fifty years, the Protestant emphasis on spiritual journeying has become secularised and found renewed expression in New Age, or "alternative", spirituality, which embraces notions of self-discovery and self-improvement and routinely uses the word "journey" to describe an individual's emotional or spiritual development (Heelas and Woodhead 2005, pp. 5–8). In the late twentieth century, the idea of combining spiritual questing with physical pilgrimage took shape around the Camino de Santiago in northern Spain, partly thanks to New Age gurus Paulo Coelho and Shirley MacLaine, whose respective best-selling pilgrim narratives did much to promote the spiritual potential of the challenging pilgrimage walk (Chemin 2012). Today, the most popular Camino guide, written by the late John Brierley (2023)—and subtitled "a practical and mystical manual"—includes sections headed "the mystical path" and "personal reflections" and owes much to Coelho's New Age vision. Coined a "therapy route" by Nancy Frey (1998), the Camino has become a magnet for those seeking emotional and spiritual healing as well as those escaping modernity through a practice deemed to be anchored in an authentic past (Egan 2010; Nilsson and Tesfahuney 2016; Mikaelsson 2012; Genoni 2010; Slavin 2003).

In Britain, the success of the Camino, the well-publicised health benefits of outdoors exercise, and a renewed nostalgia for the countryside (inspired by popular authors such as Robert Macfarlane) have all contributed to the walking pilgrimage revival. Encouraged by ventures such as the British "Sacred Land Project", which aimed to re-establish "ancient ways which have been trodden for centuries" (Palmer and Palmer 1997, p. 91) and, more recently, by those of the British Pilgrimage Trust (https://britishpilgrimage.org, accessed on 3 March 2020), Christian churches and local tourism organisations have been eager to capitalise on the new interest in pilgrimage (Williams 2020; Riley 2022). Pilgrimage, for example, is recognised as a way of revitalising Britian's ailing churches (Rutman 2023).

Throughout Europe, the Camino has become a model for a plethora of pilgrimage walking trails. Like the Camino, many of them claim to be following medieval pilgrimage routes, while others offer walkers a Camino-style passport and guidebooks which emphasise the spirituality of the walk, a phenomenon which Marion Bowman and Tina Sepp call "Caminoisation" (Bowman and Sepp 2019). In Britain, those with a "Celtic spirituality" bias are especially popular (Cusack 2013; Maddrell 2015) and help to strengthen the post-secular spiritual ethos that the divine is to be found in the natural landscape rather than in traditional institutional places of worship (Bailey 2023). What these newfound pilgrimage trails also do, of course, is to prioritise the journey over the destination in the popular imagination and reinforce the idea that pilgrimage necessitates walking long distances on foot (LeSueur 2018).

### 3. The Pilgrimage Journey

Trends in scholarship and Western culture have, then, given increased prominence to the pilgrimage journey in recent years, with the Camino doing much to propagate the idea that long-distance walking to a Catholic shrine is normative as well as a traditional way of gaining spiritual merit. However, until very recently, few European Catholic pilgrimages were undertaken by foot if faster and more convenient alternatives were available. Pilgrims heading to the shrine of Our Lady of Lourdes in south-western France, for example, piled onto trains the moment steam transport was invented and quickly moved on to buses and

then to car travel in later years (Eade 2015). Although the twentieth century saw the gradual increase in Roman Catholic and Anglo-Catholic walking pilgrimages as special events—the annual Chartres pilgrimage and Pilgrim Cross walk being prominent examples—long distance hiking was not a particularly common feature of Christian pilgrimage, particularly in western Europe.

The pilgrimage to Santiago was no exception in this respect. Until the 1970s, the practice meant, "getting to the Cathedral by any means necessary, most often by car, bus or train" (Sánchez y Sánchez and Hesp 2016, p. 3), and the Camino itself was originally planned along highways for motorists (Talbot 2016). In more recent years, low-cost flights and airports at Santiago, Jerusalem, Rome, Knock, and Lourdes have made journeys to sacred shrines even faster and easier. Christian pilgrimage worldwide is frequently characterised by Pullman bus tours, private minibus groups, and air travel and not necessarily by footsore, long-distance walkers.

In the Middle Ages, many Christian pilgrims similarly sought ease, efficiency, and speed whenever they could. This is not to argue that walking to a pilgrimage destination was held as worthless. Medieval hagiography is full of saintly pilgrims who walked long distances, and contemporary churchmen urged all pilgrims to emulate this saintly example. A twelfth-century sermon, the *Veneranda Dies*, recommended that pilgrims should travel to St James's shrine at Santiago in poverty and go barefoot like the Apostles (Veneranda Dies 1996, p. 28).

These recommendations were certainly met by some pilgrims: we find accounts of men and women who could evidently afford transport yet who trudged the pilgrimage routes by foot, undertaking brutalising travel regimes for the sake of their souls. Although some had pilgrimage imposed on them as a judicial sentence, others voluntarily exposed themselves to strenuous travel as a form of penitence. Examples include the countess who reportedly walked "bare foot" (*nudis pedibus*) to St Thomas's shrine at Canterbury (Benedict 1876–1885, p. 257) and the lame man, "despising comfortable travel by carriage" (*vehiculi solatio contempto*) (p. 147).

However, evidence suggests that these kinds of harsh, penitential journeys were not always the norm for ordinary layfolk travelling of their own volition. The two Canterbury cases were singled out as extraordinary by the hagiographer who recorded them, just as in modern Catholicism, where penitential walking might be accorded "celebrity status" (King 2005, pp. 61–62). As this article will show, travel by carriage, cart or on horseback was often the preferred mode of transport for those with financial means. Notwithstanding the ideology of the humble, penitent pilgrim idealised by Christian sermon writers, many medieval pilgrims were intent on reaching their destination as efficiently, quickly, and safely as possible.

Pilgrims travelling to the shrine of St James at Santiago are a case in point, as might be illustrated by the twelfth-century *Liber Sancti Jacobi*. The *Liber*—otherwise known by its manuscript title, the *Codex Calixtinus*—includes, among its five components, the *Veneranda Dies* and the *Pilgrim's Guide*. Apparently ignoring the advice issued in the *Veneranda Dies* that pilgrims should travel in poverty and walk unshod in imitation of the Apostles, the author of the *Pilgrim's Guide* seems to have travelled by horseback and assumes that his readers would do the same (The Pilgrim's Guide 1993, pp. 86, 88, 89, 93). A third text in the *Liber*, a collection of twenty-two miracle stories, indicates that riding was the expected norm for all but the very poorest. In instances where a method of travel is mentioned, most pilgrims are depicted on horseback, and those who are forced to walk are shown to be severely—and unfairly—disadvantaged (The Miracles of Saint James 1996, pp. 70–72, 80–81, 81–83, 94, 95).

Although the *Pilgrim's Guide* seems to anticipate a pilgrim readership, the text does little to suggest that the travel experience was itself considered spiritual. The author of the *Guide* warns about the challenges pilgrims might meet on their journey, such as poisonous fish, overloaded ferries, and disagreeable locals, but these potential menaces are not framed within any kind of religious or spiritual discourse (The Pilgrim's Guide 1993, pp. 88–89, 89,

91, 92–93, 94–96). Moreover, less than a quarter of the text focuses on the journey, whereas three quarters is dedicated to the various churches found along the way. The longest section describes the cathedral at Santiago at the journey's end. In many ways, the *Pilgrim's Guide* functions more as a church guide than as a route guide, and its religious considerations are reserved for pilgrim destinations rather than focusing on the journey. Unlike John Brierley's modern *Pilgrim's Guide*, there are no sections offering mystical ponderings and personal reflection.

We might question the spiritual importance of pilgrimage travel further by considering one of the most famous long-distance pilgrims of the Christian Middle Ages: Margery Kempe. Born in about 1373, Margery was a housewife from Bishop's Lynn in Norfolk, England, who dedicated a large part of her adult life to travelling to Rome, Santiago, and the Holy Land, as well as to numerous other holy places in England and northern Europe. In her later years, Margery committed her religious experiences to writing, and her Middle English autobiography, *The Book of Margery Kempe*, serves to strengthen the possibility that some lay people attached little spiritual importance to religious travel. Margery, who was deeply devout, was renowned for her exuberant public displays of piety. Yet, as will be shown, her approach to travel—like that of the *Pilgrim's Guide* author—seems to be entirely pragmatic.

Margery's long-distance travels might be divided chronologically into two periods: the first taking place when she was in her forties and the second when she was in her sixties, covering the years 1413–1417 and 1433, respectively. In her earlier travels, when Margery was relatively young and fit, her mode of transport seems to be of little importance to her, and she only refers to it when it falls outside her everyday experience, as in the case of sea voyages or mule transport in the Holy Land. If she otherwise walked, we can presume that she considered walking unremarkable.

During Margery's later travels, readers see her paying for rides in a number of pilgrim wagons, hiring a horse for riding and trying unsuccessfully to beg a lift in a carriage with a well-to-do widow (Kempe 2004, pp. 278, 279, 283–84, 287, 282). Her journeys are often tough: she is forced to sleep in a barn, she travels with vermin-infested beggars, and she falls sick after struggling to keep up with her fitter companions (pp. 278, 281, 285). However, these austerities are forced upon her through necessity, and Margery never gives her travel hardships any religious significance. Her main concerns are practical rather than spiritual, and her narrative suggests that constant anxieties about her physical safety surface foremost in her mind (pp. 275, 277, 278, 281, 285).

One of Margery's pilgrimages, which particularly stands out in this respect, is that to the shrine of St James at Santiago in 1413 (pp. 147–48). Remarkably, Margery manages the return trip from England in twenty-five days, less time than it takes modern pilgrims to walk one way along the Camino Francés from the customary start point at St-Jean-Pied-du-Port. The outward leg of Margery's pilgrimage took her a mere seven days. This astonishing turn of speed was due to the opening of a direct sea route between Bristol and the Spanish port of La Coruña, forty miles north of Santiago. As explained by Wendy Childs, this late-medieval travel innovation cut weeks, even months, from the overland journey and was far less financially burdensome (Childs 1999). It made the overland route—the modern Camino Francés—largely obsolete for those coming from the British Isles. Given Margery's propensity for penitential behaviour, we can only assume that this short-cut to Santiago was not thought of as any less commendable than travelling the longer way round.

If we look at evidence for that most gruelling of all medieval Christian pilgrimages—that to the Holy Land—there is often a similar low-key approach to the travel element in autobiographical accounts. One pilgrim, the fifteenth-century English physician, Richard of Lincoln, does not appear to have envisaged his journey across Europe as a pilgrimage at all because the word "pilgrimage" only appears in his text when he reaches his destination. Here, the term is restricted to a handful of rubricated sub-headings under which he describes short circular tours undertaken in and around Jerusalem. These localised "pilgrimages"—which

include "Pilgrimages in the city of Jerusalem" and "Pilgrimage to Mount Syon"—suggest that "pilgrimage", in this context, was understood as a religious touring itinerary within specific geographical boundaries (Richard of Lincoln 2013, pp. 29, 32, 34–36). As will be discussed, the concept of pilgrimage as a relatively short destination-based tour was by no means uncommon in this period.

In general, then, a number of key sources from the twelfth to the fifteenth centuries suggest that the journey was not always deemed the most significant element of pilgrimage in the Middle Ages. Although longer pilgrimage journeys from Europe tended to be punctuated by visits to places of religious interest, the experience of traveling between them was not always perceived as spiritual by those undertaking these journeys. Many of the travel adventures narrated by pilgrim authors are characterised by their secular, rather than by their spiritual, content. When the German Dominican preacher, Felix Fabri, travelled to the Holy Land for the second time in April 1483, the sea voyage made such an impression on him that he was prompted to craft a lengthy treatise. However, this was not a devotional tract but a technical discourse on the properties of different Mediterranean galleys. As will be shown, it was only when Fabri arrived at his destination that he turned his thoughts to serious religious matters and the salvation of his soul.

## 4. Destination Journeying

For Felix Fabri, Margery Kempe, and many other long-distance pilgrims in the Middle Ages, the religious aspect of pilgrimage—and the meaningful physical and spiritual hard work—only really seems to materialise on arrival at their destination. It was Margery's devotional and penitential activities at the holy places she visited which were spiritually efficacious, not the effort of getting there. In contrast to her fast route-marches across Europe, she lingered for weeks on end in Rome, undertaking the prescribed holy circuits, living in penury, and carrying out tough penances (Kempe 2004, pp. 116–35). These are self-imposed austerities which she does not appear to have voluntarily embraced while travelling.

A similar scenario can be seen with respect to Felix Fabri in Jerusalem. Upon arrival, he and his fellow pilgrims are taken on a tour of the city's holy places. This transpires to be a gruelling devotional journey necessitating physical, mental, and spiritual endurance:

> Indeed, no one should think visiting the holy places to be a light task. There is the intense heat of the sun, the walking from place to place, the kneeling, and prostration. Above all, there is the strain which everyone puts on himself in striving with all his might to rouse himself to earnest piety and comprehension of what is shown him in the holy places, and to devout prayer and meditation, all of which cannot be done without great fatigue, because to do them fitly a man should be at rest and not rambling about. To struggle after mental abstraction, whilst bodily walking from place to place, is exceeding toilsome. Some of our pilgrims were unable to undergo it and went down to the hospital to rest, so that it was only less than half of them that persevered in the labour of pilgrimage (Fabri 1892–1893, p. 299).

Fabri's experience was not unique. Visiting Jerusalem four centuries earlier, Richard of St Vannes reportedly, "wore down his body with fasts, never without tears, never without prayers" at each place he visited (Hugh of Flavigny 2001, p. 40). For the pious, visiting Jerusalem was an exercise of physical, emotional, and spiritual stamina.

Holy cities, such as Jerusalem and Rome, were saturated with sacred places demanding a pilgrim's attention, and the most significant were often linked by prescribed routes taking pilgrims on mini-journeys from place to place. Rome and Jerusalem, for example, had approved "Stations", where pilgrims would halt and spend time in prayer. In medieval Jerusalem, the Stations clustered around the Holy Sepulchre Church and would later evolve into the *Via Dolorosa*, the "Way of Sorrows". In Rome, the "Stations" referred to by Margery Kempe were based on seven principal churches, but the most ambitious pilgrims attempted the rounds of many more. William Wey, a fifteenth-century Devonshire priest, claimed

to have counted 467 churches worthy of a pilgrim's visit. He lists by name well over a hundred, indicating that, for ardent Christians, visiting Rome could be a time-consuming business (Wey 2010, pp. 191–99).

Destination-based pilgrimages were also possible within the narrower confines of single churches in the Middle Ages. Although the Stations of the Cross would not become a regular feature of Catholic churches until after the Middle Ages, the ritual movement of pilgrims through sacred space was common in major churches owing to the proliferation of relics, shrines, and altars. According to William Wey, Rome's principal church, St Peter's, possessed 105 altars, each containing a holy relic (Wey 2010, p. 192).

From around the twelfth century, destination-based rituals grew in importance due to the development of indulgences (Swanson 2007). Granted by the Pope and other bishops to pilgrims visiting specific shrines, indulgences promised remission of time spent in purgatory. Rome and the Holy Land were particularly rich in indulgences. William Wey's list of Roman churches includes information about those available at each one, while Felix Fabri and Richard of Lincoln penned crosses in their manuscripts to indicate each Holy Land indulgence they obtained. The importance given to indulgences in these texts indicates that the accumulation of these much-desired spiritual benefits must have been a major pilgrimage motivation.

Significantly, indulgences were usually attached to specific places, not to journeys; with a few exceptions, pilgrims were required to be physically present at their destination to receive the benefit. Moreover, the availability of multiple indulgences at some shrines accentuated the importance of destination rituals and encouraged extended stays. Although pilgrims arriving at Santiago were offered a "basic" grant of the remission of one-third of all sins, "extras" could be added by reappearing in the cathedral at specific times or for particular occasions (Wey 2010, p. 216). The temptation to linger for a protracted time at a shrine, for the purpose of accumulating as many indulgences as possible, must have been great.

The idea that pilgrimage took place at one's destination within the bounds of a sacred site is one that many modern Catholics would readily understand. The plenary indulgences offered at Santiago are still wholly contingent on pilgrims' presence in the cathedral, for instance (Frey 1998, p. 22). Prolonged visits are also a recognised feature of many contemporary pilgrimages. One example is the shrine of Our Lady at Lourdes, established in the mid nineteenth century following a series of Marian apparitions claimed by the visionary Bernadette Soubirous. Although there has never been a tradition of walking to Lourdes, modern pilgrimages are no less physically exacting. Today, the standard Lourdes pilgrimage is undertaken during five or six days and comprises arduous, daily rounds of ritual movement and processions within the sacred grounds. A pilgrim's diary of activities typically includes up to five set events, beginning with a mass at eight-thirty in the morning and ending with a daily torchlight procession at nine o'clock in the evening (Thomas et al. 2018, p. 414).

It is Ireland, however, which hosts possibly one of Catholicism's toughest pilgrimages. The country is endowed with possibly hundreds of sacred sites, often holy wells, where pilgrimage (*turas*) typically involve "doing the rounds" at the destination, as exemplified at Glencolmcille in County Donegal (Jackman 2023). Although these rituals might take hours to complete, at Station Island at Lough Derg, Country Donegal, the pilgrimage lasts three days (Hedderman 2022). During the pilgrimage season, participants arrive by boat and, over the three days, complete repetitive Stations around the penitential "beds", involving a continuous cycle of prayers, ritual gestures, and kneeling. The pilgrimage is undertaken bare foot, and other austerities include all-night vigils and fasting.

The point about Lourdes and Lough Derg is that, in each case, the formal pilgrimage rituals only commence on arrival at the sacred location. These are not isolated examples; numerous other Catholic pilgrimages around the world place great emphasis on destination rituals, which pilgrims often perform in various penitential ways, perhaps going barefoot,

on their knees, or crawling. In other words, although penitential walking is a feature of many Catholic pilgrimages, it is often done within the confined space of the site itself.

## 5. Crossing into Sacred Space

One consequence of location-bound pilgrimage—whether the sacred site is a church building, a shrine complex, or a holy city—is the symbolic existence of an imaginary line which pilgrims cross as they move from their journey into sacred space (Gesler 1996, p. 102). In many cultures, the physical movement of pilgrims from the secular to the sacred is signified by a change in the pilgrim's deportment, dress, and behaviour. In her study of the Marian cult on the Greek island of Tinos, Jill Dubisch describes how pilgrims approaching the shrine cross themselves, remove their shoes, fall to their knees, and undergo austere practices, such as crawling to the church (Dubisch 1995, p. 23). In some circumstances, pilgrims arriving at a shrine might more radically alter their appearance as a symbol of humility and personal sacrifice (Greenfield and Cavalcante 2005).

Medieval Christians often made similar preparations, removing their shoes and assuming a penitential attitude at some customary point a short distance from the shrine. Chapels or churches frequently marked these boundary locations. A final staging post, represented by a church, was established on Mount Joy, two miles from Jerusalem, where pilgrims had their first glimpse of the holy city. The twelfth-century pilgrim, Theodoric, explained that pilgrims reaching this point would attach crosses to their clothes, remove their shoes, and switch to a humbler demeanour (Theodoric 1988, p. 310). Daniel, a Russian abbot, added that pilgrims dismounted at Mount Joy and continued to the city on foot (Daniel 1988, p. 127).

Other pilgrim routes had their own "mount joy". For Santiago-bound pilgrims, this was Monte de Gozo, and for those approaching Bury in Suffolk, it was the last mile, when the bell tower of the abbey came into sight (Pinner 2015, p. 115). For Canterbury pilgrims, it was the vantage point of Harbledown, where the cathedral was first glimpsed and where, famously, in 1170, Henry II dismounted from his horse and walked barefoot into the town to atone for his part in Becket's murder (Urry 1999, pp. 159–60). Walking the final mile to the shrine may have been common practice: a late twelfth-century miracle story describes how the wife of a knight riding from Pontefract dismounted at this spot and walked the final leg of her journey (Benedict 1876–1885, p. 233). Significantly, these cases of penitential walking were not private, long-distance tests of endurance, but rather were symbolic public acts performed within sight of a sacred destination.

For many ordinary pilgrims, then, penitential journeying did not mean a long, challenging walk but was restricted to areas at, or near to, a demarcated holy place. Medieval examples are particularly striking in this respect. Despite attempts by medieval churchmen to endow pilgrimage journeys with Christian meaning, pilgrims like Margery Kempe, Felix Fabri, William Wey, and Richard of Lincoln do not appear to have envisaged their travel experiences within a spiritual framework. For these pilgrims at least, the main event took place at their destination, a pattern that is still characteristic of many Catholic pilgrimages today.

## 6. Journeys: Secular or Devotional?

For the past half century, the social sciences have been engaged in an ongoing debate concerning the relationship between the conceptual categories of pilgrimage and tourism.[2] The debate predominantly revolves around how "religious" or "secular" each practice is deemed to be. The current academic position has largely moved away from an either/or model and now holds that, in the modern world, it is increasingly difficult to distinguish between pilgrimage and tourism insofar as religious and secular practices often merge (Badone and Roseman 2004; Collins-Kreiner 2018). It has been argued, for example, that pilgrims and tourists frequently switch between roles or that pilgrims incorporate a mixture of secular and sacred elements in their pilgrimages (Turner and Turner 1978, p. 20; Strausberg 2011, pp. 65, 88, 90). Another popular idea is that tourists and pilgrims sometimes

undergo a transformative experience and "convert" from one identity to the other (Frey 1998, pp. 135, 197; Winter and Gasson 1996, p. 178). One variant of this role-switching theory, which has been hinted at in this article, is the possibility that some pilgrims leave their points of departure as "tourists" and remain secular travellers until arrival at their destination, when social convention, rather than religious conversion, prompts a dramatic change in behaviour.

Although the World Tourism Organisation recognises pilgrimage as a subset of tourism, many academic theorists find it difficult to break away from the structural model which sets tourism and pilgrimage against one another as oppositional categories (LeSueur 2018; Margry 2014; Vukonic 2005, p. 244). In this binary model, tourists are given worldly, material attributes, while pilgrims are provided with contrasting spiritual ones (Strausberg 2011, p. 20). Tourism—often associated with consumerism and pleasure-seeking—easily accrues negative values in this structural imagining. In *Pilgrim Stories*, Nancy Frey describes how this viewpoint is disseminated by modern pilgrims on the Camino. Walkers self-identifying as "pilgrims" distinguish themselves as "true" pilgrims by setting themselves apart from those they consider to be mere superficial tourists. "Real" pilgrims are assumed to be those who travel slowly for a prolonged period of time, carry their own rucksacks, live frugally, only stay at pilgrim hostels, and endure many other challenging—but ultimately self-improving—discomforts. Motorised transport, luggage transfer, expensive hotels, and fine dining are an anathema to these self-styled "true" pilgrims, these being considered the mark of the shallow, frivolous tourist (Frey 1998, pp. 50–51, 125–36). In this reading, the tourist becomes a hedonistic, self-indulgent consumer of worldly pleasures: a kind of anti-pilgrim giving pilgrimage a bad name.

The tourist/pilgrim dichotomy, which pits the secular pleasure-seeker unfavourably against the spiritual truth-seeker, is not a new phenomenon. Neither is the idea, also held by many Camino pilgrims, that the model for authentic pilgrimage can be found in the distant past. Although terms such as "tourist" and "secular" might be argued as being anachronistic with respect to the Middle Ages, the notion that present-day society creates an irreligious breed of "false" pilgrims bent on travelling for the sole purpose of enjoyment was a common theme in late-medieval polemics. Satirists and religious commentators like William Langland, author of *The Vision of Piers Plowman*, denounced the pleasure-seeking, secular pastime that—they claimed—pilgrimage had become.

A famous text critiquing late-medieval pilgrimage is the testimony of William Thorpe in 1407, which depicts the pilgrimage journey as an uproarious cavalcade of irreverent merriment involving the singing of "wanton songes", the loud playing of musical instruments, and the "jangling of Canterbury bells". Pilgrimage, it claims, was more about fleshy delights and extravagant money-spending than about spiritual concerns (*Acts and Monuments* 1853–1870, p. 268). Other late-medieval commentators emphasise the untoward consumerism and hedonistic excess said to be tarnishing the practice of pilgrimage. Most of these criticisms focus on the journey. In 1529, Thomas More referred to pilgrims as, "gadding about idly, uncontrollably, with revelling and ribaldry, gluttony, wantonness, waste and lechery" (More [1529] 1981, p. 226). Pilgrimage, he observed, even turned the most respectable of wives into impudent hussies, such as the country women who unashamedly accompanied their journeys with raucous, ribald, and "unwomanly songs" (236).

Although it is tempting to view these denunciations of false pilgrims as an indication of moral decline, it may well have been the case that much pilgrimage travel undertaken by the laity always had an element of irreligiosity about it and that the increasing tendency to condone "touristic" behaviour on pilgrimage in the late Middle Ages signalled a narrowing of attitudes by conservative churchmen rather than a change in popular practice. Feast days—which often drew in crowds of visitors from far afield—were, after all, renowned for attracting boisterous and drunken behaviour throughout the Middle Ages, and it is not unlikely that this festive spirit was shared by pilgrims. It is suggestive, for example, that the *Veneranda Dies* finds it necessary to caution prospective pilgrims against debauchery and

drunkenness on the Camino and includes a long homily on the evils of excessive drink and brawling (Veneranda Dies 1996, pp. 30, 32–34). Interestingly, one Jerusalem pilgrim, Abbot Daniel, admits to travelling to the Holy Land, "with every kind of sloth and weakness, in drunkenness and doing every kind of unworthy deed", behaviour for which he makes full repartition once at his destination (Daniel 1988, pp. 120–21).

The festive, "touristic" characteristic of pilgrimage journeying remains a well-observed feature of many modern Catholic pilgrimages. One example is documented by an anthropologist who joined a group of Portuguese villagers travelling to the shrine of Our Lady of Fatima in 2004. Some of the pilgrims, for penitential reasons, had vowed to go by foot, but the majority travelled in family or neighbourhood groups by bus or car. They called their trip an excursion (*excursado*) and the day-long journey involved sight-seeing detours, stops for picnic lunches, plenty of wine-drinking, joke-telling, and the singing of popular folk songs. The organiser, we learn, was rarely without a bottle of wine in her hand (Gemzöe 2005, pp. 35–37).

This twenty-first century journey is reminiscent of Chaucer's fourteenth-century *Canterbury Tales*, in which a group of fictional pilgrims heading for St Thomas's shrine at Canterbury tend towards the tourist side of the tourism/pilgrimage equation. They ride on horseback, patronise comfortable lodgings, tell bawdy stories, and generally have a good time. Frustratingly, we are not privy to what happens to Chaucer's pilgrims once they arrive at Canterbury. However, it is possible to follow the modern Portuguese pilgrims to their destination and discover that what takes place on their arrival is a spectacular turn-about. When these fun-loving Catholics arrive at the shrine of Our Lady, a dramatic transformation occurs. One pilgrim changes out of blue jeans into black leggings in readiness to fulfil her vow of walking to the shrine on her knees; and, inside the sanctuary, lines of women are seen walking, crawling, praying, and crying. The switch from tourist to pilgrim—from drunken merriment to penitential sobriety—is abrupt, ritualised, and deeply emotional (Gemzöe 2005, pp. 37–38).

Although it seems likely that festival revelries were also a common feature of historical pilgrimages, they only tend to surface in the historical record in negative contexts. Censorious comments made by medieval commentators about socially disruptive behaviour on feast days are certainly noted by historians, but such occurrences are not often recognised as part of the pilgrimage experience perhaps, in part, because scholars tend to view medieval pilgrimage as a serious, pious undertaking rather than as a drunken party. From a modern perspective—and particularly that shaped by Protestant culture—there is something slightly sacrilegious about hedonistic behaviour on pilgrimage. As we have seen with respect to the modern Camino, fun-loving pilgrims suggest a pure form of religiosity gone worrisomely "bad".

However, there is also the possibility that pilgrimage for the masses always had included this mixture of the irreligious and the religious and that the notion of the spiritual journey was often more an ideology—and clerical wishful thinking—than a social reality. It is also a mistake to assume, as did medieval critics, that the profane nature of a pilgrim's journey indicated a lack of religious commitment. On the contrary, modern case studies show that swinging from one pole of excess to the other—from holiday to holy day—only seems to heighten pilgrims' emotions and intensify their spiritual experiences at the shrine.

Although this article has focused on Christian and post-Christian examples, it should also be noted that other faiths and cultures often give a similar pre-eminence to pilgrimage destinations. As at Lough Derg and other Catholic shrines, many involve circular movement and repetitive rounds of ritual and prayer at delineated sacred sites. The five-day Hajj, undertaken by Muslims at and around Mecca, is the obvious example, but Hindu, Buddhist, and Shinto pilgrimages also feature circumambulation rituals at sacred sites. The Shikoku pilgrimage in Japan can take six weeks to walk, involving visiting eighty-eight temples. In instances where on-site pilgrimage practices are physically challenging, the linear journey from home to the pilgrimage destination is usually credited with little or no

spiritual worth, and journeys are made as speedily, as efficiently, and as comfortably as possible (Reader 2013).

## 7. Conclusions

This article has come about, in part, from my own experiences of talking to pilgrims and, in particular, by a conversion with a student who claimed that the Lourdes pilgrimage was not a "proper" pilgrimage because it did not involve walking to the shrine. This understanding of pilgrimage as "a journey on foot" (British Pilgrimage Trust, https://britishpilgrimage.org, accessed on 3 March 2020) is becoming increasingly widespread. At the time of writing, a new "Jerusalem Way" is being established in Israel, a 450 km hiking trail which includes a section adapting the ancient route from the old port of Jaffa to Jerusalem (Way to Jerusalem), notwithstanding the fact that historical pilgrims like Felix Fabri and William Wey undertook this journey by donkey and, most certainly, not on foot.

As this article has shown, the method of travelling to a shrine was often immaterial to medieval pilgrims, as literally exemplified in virtual pilgrimage where pilgrims travelled spiritually, but not physically, to the Holy Land (Beebe 2015; Donkin and Vorholt 2012; Hillman and Tingle 2019). In many ways, the realities of medieval pilgrimage were much more varied than is often realised today. There were, as there are now, pilgrims who—for penitential reasons—made the journey to a saint's shrine as mortifyingly painful as possible. There were others who had no option but to go on foot, and there were some shrines and occasions where walking may have been customary. However, there were also pilgrims for whom the act of moving through secular landscapes—between holy places—does not appear to have been particularly imbued with spiritual significance. Despite attempts by the medieval Church to sacralise religious travel and give it interior meaning, the serious spiritual journey only began for these pilgrims once they reached their destination.

Until very recently, little changed for the world's practising Roman Catholics. Nonetheless, Catholicism is not completely resistant to cultural change. *The Catholic Herald* introduced its first organised walking pilgrimage in 2019 (Cash 2020), and walking trails commemorating Catholic saints are fast appearing, such as the *Via di Francesco* (St Francis Way), tracing the footsteps of St Francis from Assisi to Rome. In Ireland, heritage walking trails now feed onto the pilgrimage site at Croagh Patrick, and for those unwilling to undergo the gruelling three-day pilgrimage on Station Island, there is a recently established pilgrim path which skirts around Lough Derg (Lough Derg 2022). Catholics, of course, also number among the pilgrims who walk the Camino each year. Perhaps most notably, the last year has seen the launch of the Walsingham Camino (Official Launch 2022) and a project to establish pilgrim trails across each Catholic diocese in England and Wales (Hearts in Search of God). The concept of pilgrimage is changing to fit in with a changing world.

As for the spiritual makeover given to the modern Camino, there is something of an irony in the enthusiastic espousal of a journeying ideology which, in the Middle Ages, was vigorously promoted by the Church and yet was so often disregarded by the pilgrims themselves. Had the twelfth-century author of the *Veneranda Dies* been able to time-travel into the twenty-first century, his puzzlement at the cursory visits made by pilgrims to St James's shrine would, no doubt, have been countered by his joy at seeing so many men and women travelling light, and on foot, in imitation of the Apostles.

**Funding:** This research received no external funding.

**Data Availability Statement:** Not applicable.

**Conflicts of Interest:** The author declares no conflict of interest.

## Notes

1   Other forms of medieval "pilgrimage" (*peregrinatio*), such as crusading, judicial pilgrimage, and virtual pilgrimage, are not covered in this article.

2   The progenitors of the debate are traditionally said to be Boorstin (1961), MacCannell (1976), and Graburn (1978).

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
