# Peer review of "Journey or Destination? Rethinking Pilgrimage in the Western Tradition"

_religions, doi:10.3390/rel14091157_

Round 1
Reviewer 1 Report
This is a well written and convincingly argued paper that clearly explains how the emphasis on "difficult" and "transformative" journeys as a part of "real" pilgrimage experience could be more an idea that does not work entirely in practice.
I do not have any particular suggestions to the author, except one that I strongly recommend. Namely, it is clear that almost all references, case studies and authors personal experience are based on Catholic pilgrimage sites and routes in Europe. And, the author's arguments work within this context, however, things become more complex if we are talking about the idea of pilgrimage in general. See for example recent study by Reader and Shultz on Japanese pilgrims in Shikoku (2021). My suggestion is that the author clearly states that the paper deals with Catholic pilgrimages. Starting from the title.
Author Response
The article focuses on pilgrimages from a Christian and post-Christian tradition in western society (i.e. not only Catholic pilgrimages but also Anglican, Protestant, and post-secular ones). However, the reviewer is correct in pointing out that I should have made it clearer that the article only focuses on one cultural iteration of pilgrimage and not on pilgrimage more generally. In response to this valid criticism, I’ve amended the subtitle to read, ‘in the Western tradition’ and have added ‘pilgrimage in Christian and post-Christian society’ to the abstract to make this absolutely clear. ‘Christianity’ is also added as a keyword. (Changes highlighted in blue in the MS.)
Reviewer 2 Report
Excellent work

Author Response
Thank you. That's very kind.
